# Comparison of β-Glucan Content in Milled Rice, Rice Husk and Rice Bran from Rice Cultivars Grown in Different Locations of Thailand and the Relationship between β-Glucan and Amylose Contents

**DOI:** 10.3390/molecules26216368

**Published:** 2021-10-21

**Authors:** Pattraporn Phuwadolpaisarn

**Affiliations:** Department of Chemistry, Faculty of Science, Chandrakasem Rajabhat University, 39/1 Chatuchak, Bangkok 10900, Thailand; pattraporn.p@chandra.ac.th

**Keywords:** β-glucan content, milled rice, rice husk, rice bran, amylose content

## Abstract

β-glucan is a dietary fiber that is beneficial to human health, and its content varies according to its different parts, type of cereal grain, and growing environment. In this study, the β-glucan of milled rice, rice husk, and rice bran fractions, as well as the amylose content of milled rice fraction, from 38 selected rice-paddy grains from six regions of Thailand were quantitatively determined. The milled rice of the Sakon Nakhon (SN) cultivar grown in the northeast contained the highest β-glucan content (0.88 ± 0.03%), followed by the milled rice of the Jow Khao Chiangmai (JKC) cultivar (0.71 ± 0.03%) and rice bran of the Sew Mae Jan (SMJ) cultivar (0.67 ± 0.03%) grown in the north. The results reveal that the rice cultivars from each region showing variation in the β-glucan level in each fraction, which is mainly found in milled rice and rice bran, are similar to those found in other cereal grains, although low amounts are found in the husk. The amylose and β-glucan contents in the milled rice fraction showed a strong negative correlation (r = −0.805; *p* < 0.0001). This new information about the β-glucan content of Thai rice cultivars could be used for the development of cereal-based functional food.

## 1. Introduction

Thailand is an agricultural country that produces rice as its main staple food, which is grown on approximately 10 million ha of land (about 20% of the total land area of Thailand) [1]. Thailand is the world’s sixth-largest rice producer, of which paddy rice and milled rice account for about 27.2 and 18 million tons, respectively [2]. After milling, paddy rice is separated into four parts, namely, milled rice (endosperm), rice bran, rice germ, and rice husk, each having different chemical compositions, phytochemicals, antioxidant activities, and applications [3,4,5,6]. Milled rice is mainly consumed by humans for nutrition and is also particularly valuable for use in functional foods. The waste fractions from rice milling comprise bran and husk that are of low economic value but have high nutritional content. Rice bran is commonly a mixture of bran and germ (embryo) that contains high levels of several phytochemicals that have antioxidant properties; therefore, they are extensively utilized in functional food and as food additives [3,5,7]. Rice husks have an antioxidant defense system that protects the rice seed from oxidative stress, and although they have valuable nutritional properties, they are inedible [5,8,9]. In Thailand, rice bran is mostly used as animal feed and a source of rice bran oil and bioactive phytochemicals, such as γ-oryzanols, that are found in the oil phase [3,5,10]. However, the bran is generally not fully reused, and the utilization of the husks is limited. Therefore, they are usually discarded as waste or eliminated by burning [8].

Thailand has 138 varieties of rice that are classified based on the ecosystem of the riceland on which they were cultivated and their photoperiod sensitivity and cooked texture [11]. There are four ecosystems for rice cultivation: lowland rice, upland rice, floating rice, and deep-water rice. Lowland rice is cultivated in almost every region of the country. Rainfed lowland rice is generally grown in the northern and northeastern regions, whereas irrigated lowland rice is mostly grown in the southern region and on the central plain [12]. Floating and deep-water rice are grown in some eastern provinces and in the central plain, such as in Prachinburi and Ayutthaya, respectively.

Based on the flowering response to light, rice cultivars are categorized into photosensitive and non-photosensitive; accordingly, the cultivars’ planting times are different [13]. Photoperiod-sensitive rice cultivars are always grown in the rainy season and are sensitive to winter flowering, while photoperiod-insensitive rice cultivars can be grown in the off-season [11,14]. 

Amylose is a major component of rice and strongly influences its cooked texture. Glutinous rice and non-glutinous rice are classified based on their amylose/amylopectin ratio: glutinous rice has low (<5%) amylose in its starch but is high in amylopectin [15]. Finally, rice is classified by the percentage of amylose in starchy endosperm as waxy (0–5%), very low (>5–12%), low (>12–20%), intermediate (>20–25% amylose), and high (>25–33%), as suggested by Juliano [16]. In contrast, the amylose content in commercial rice is classified as low (<20%), medium (21–25%), and high (26–33%) [17].

β-glucan is a significant component of the cell walls of higher plants that belong to the grass family (Poaceae), such as rice, wheat, barley, and oat; therefore, it is a common component of many human diets and feed formulations [18]. Structurally, β-glucans are usually characterized by mixed-linkage (1–3),(1–4)-β-d-glucan (hereafter referred to as “β-glucan”) and are found as a chain of linear homopolysaccharide glucopyranosyl residues linked by several (most often two or three) consecutive (1–4)-β-d-linkages separated by a single (1–3)-β-d-linkage [19,20]. The consumption of cereal β-glucan results in powerful health benefits, including moderate glycemic response, lower cholesterol, enhanced satiety after ingestion, and stimulation of healthy gut microflora [21,22,23]. Cereals containing β-glucan are widely available as a functional food offered by different areas of the food industry [24,25,26]. 

In general, β-glucan in cereal is found in the sub-aleurone layer and endosperm cell walls [27]. Different cereals show different distribution patterns in the kernels. For example, the amount of β-glucan is low in the sub-aleurone but high in the endosperm of oat kernels, is particularly found in the endosperm of barley, is mainly found in the sub-aleurone layer but scarcely in the endosperm of wheat, and is distributed throughout the grain in rye [20]. Moreover, the β-glucan in the starchy endosperm, sub-aleurone layer and aleurone layer of wheat and rice is labeled by the mixed-linkage β-glucan (MLG) antibody, as is the rice husk [28]. To date, a few studies have reported the content of β-glucan from rice, which varies from 0.4 to 0.9% [29], whereas that of non-fermented rice bran varies from 0.14 to 0.57% [30], and there are no data for husk. The β-glucan level of cereal can vary due to the nature of the genotype, growing environment, including conditions of flowering time and grain filling. Many researchers have studied oats and barley, with findings such as drier and warmer temperatures during the flowering time of barley could enhance the level of beta-glucan [31], whereas a higher temperature and lower rainfall during oat grain filling contributed to a higher level of beta-glucan [32]. Additionally, the amylose content of starchy endosperm also influences the level of β-glucan, in which the β-glucan and amylose contents of cereal grains are inversely corelated, particularly in barley [33,34,35,36].

Thailand not only has several rice cultivars distributed over entire regions but is also rich in rice milling fractions from these rice cultivars. Two major varieties are the Khao Dawk Mali 105 (KDM105) and RD6, which together cover approximately 64% of the cultivation area and have the greatest yield of paddy rice (8.1 and 4.7 million tons, respectively), so they have the most rice milling fractions compared to others [12]. These fractions are milled rice, bran, and husk, which can be a source of β-glucan. Currently, no basic information evaluating the β-glucan content of those fractions has been reported. Therefore, this research aimed to determine and compare the β-glucan content in each paddy-rice fraction (milled rice, rice bran, and rice husk) of 38 rice cultivars grown in six regions of Thailand: northern, northeastern, eastern, western, southern, and central plain. The difference in β-glucan level classification according to riceland ecosystem and photoperiod sensitivity is also discussed. In addition, to evaluate the relationship between amylose and β-glucan content, the amylose content in the milled rice fractions was determined.

## 2. Results and Discussion

### 2.1. Percentage of Each Paddy-Rice Fraction Yields

The percentage yields of rice paddy fractions of milled rice, rice husk, and rice bran, and the yield loss of all 38 cultivars are given in Table 1. The RD39 lowland cultivar of the northern region had the highest milled rice fraction (70.08 ± 1.51%) but the lowest rice husk fraction (21.28 ± 1.59%). The deep-water At1 rice cultivar of the eastern region had the highest rice bran fraction (10.00 ± 0.92%), whereas the lowland RD21 of the north contained the highest rice husk waste (29.76 ± 2.19%). The lowland JC1 cultivar of the central plain had the lowest milled rice fraction (61.16 ± 0.60%), whereas the RD47 cultivar in the south contained the lowest rice bran fraction (6.17 ± 0.47%). The yield loss to paddy-rice milling ranged from 0.88 ± 0.14 to 2.27 ± 0.91%. The yields of all fractions of rice cultivars (61.16–70.08% for milled rice, 21.28–29.76% for husk, and 6.17–10.00% for bran) were relatively close to the ideal for rice milling with a yield in the range of 61–72% milled rice, 20–30% rice husk, and 5–12% rice bran [37,38]. The yields of milled rice and its fraction of byproducts are not only dependent upon the degree of milling but also on several factors, such as varieties of rice, growing season, and environmental variability [37,39,40].

### 2.2. β-Glucan Content in Paddy-Rice Fractions

β-glucan is abundant in the cell walls of cereal grains, such as oat, barley and wheat. In rice, β-glucan is distributed throughout the grain, including in the starchy endosperm and sub-aleurone layer of milled rice, aleurone layer and maternal pericarp of the bran, as well as the husk [28,41]. In Thailand, several studies have evaluated the nutritional composition of milled rice and its byproducts, particularly rice bran [3,5,8,10,42] although, to date, data have been lacking regarding β-glucan, a soluble dietary fiber whose consumption is beneficial to human health. To address this in our study, we determined the β-glucan content of the paddy-rice fractions of 38 rice varieties using a mixed-linkage β-glucan assay kit to specifically measure the cereal β-glucan content. The β-glucan content and total β-glucan content of milled rice, rice husk, and rice bran fractions in 38 rice varieties that are cultivated in different locations of Thailand and categorized according to riceland ecosystem, photoperiod sensitivity, and texture of cooked rice are shown in Table 2.

The β-glucan distribution of each paddy-rice fraction is ranked in the following descending order: milled rice (0.11–0.88%) > rice bran (0.11–0.67%) > rice husk (0.02–0.11%). The measured contents in milled rice and rice bran are relatively similar to the findings of Demirbas [29] and Jung et al. [30], who reported a β-glucan content of rice in the range of 0.4–0.9% and that of Korean non-fermented rice bran in the range of 0.18–0.57%, respectively. Our study also presented results for the β-glucan content in rice husk which, surprisingly, had not previously been reported. The results are also supported by the finding of Palmer et al. [28], who determined the cellular location of β-glucan in whole rice grain using MLG antibody labeling detection. The β-glucan of each fraction was separately expressed as the total β-glucan content (mg) that might be found in 300 g of prepared paddy-rice grain (Table 2). Among the three paddy-rice grain fractions, milled rice had the highest total content (218.35–1749.21 mg) followed by rice bran (21.12–182.91 mg) and rice husk (12.90–81.62 mg). Among the milled rice fractions, the SN and JKC cultivars (1749.21 ± 29.62 and 1427.68 ± 56.74 mg, respectively) had much higher total β-glucan contents than the other cultivars. Similarly, the rice husk β-glucan content for SN (81.62 ± 2.36 mg) was the highest. For the rice bran fraction, however, the SMJ, DPy and SYP cultivars had a higher amount of β-glucan: 182.91 ± 1.37 mg, 157.73 ± 3.35 mg and 152.96 ± 7.08 mg, respectively.

KDM105 and RD6 had β-glucan contents in milled rice of 0.41 ± 0.02 and 0.46 ± 0.02%, respectively; rice bran of 0.20 ± 0.02 and 0.31 ± 0.01%, respectively; and rice husk of 0.06 ± 0.00 and 0.07 ± 0.00%. The total β-glucan contents in milled rice, rice husk, and rice bran of KDM105 were 842.02 ± 18.98, 46.11 ± 0.97, and 38.36 ± 0.70 mg, respectively. The total β-glucan contents in milled rice, rice husk, and rice bran of RD6 were 854.67 ± 26.14, 56.45 ± 3.97, and 79.80 ± 1.67 mg, respectively. Although these two cultivars have moderate β-glucan contents in each fraction, they are two major cultivars of Thailand. Therefore, they are another source of β-glucan content as they have a rich annual yield of rice paddy.

For each of the six regions, the highest amounts of β-glucan for selected rice cultivar fractions, in descending order, for milled rice, were found as follows: northeastern (0.88 ± 0.03% for SN) > northern (0.71 ± 0.03% for JKC) > southern (0.45 ± 0.02% for NDC4) > central plain (0.42 ± 0.02% for PT1) > eastern (0.29 ± 0.02% for At1) > western (0.21 ± 0.03% for CN1). For rice bran, the highest amounts were found in the following order: northern (0.67 ± 0.03% for SMJ) > southern (0.57 ± 0.02% for Dpy) > northeastern (0.52 ± 0.02% for SN) > central plain (0.40 ± 0.01% for JC1) = eastern region (0.40 ± 0.01% for PB2) > western region (0.39 ± 0.02% for CN1). For the most part, the rice husk showed a very low β-glucan content. In descending order, the highest levels were northeast (0.11 ± 0.01% for SN) > northern (0.09 ± 0.01% for JKC) > southern (0.08 ± 0.00% for DPy) > central plain (0.07 ± 0.01% and 0.07 ± 0.00% for JC1 and Pl2, respectively) = eastern (0.07 ± 0.00% for PB1) > western (0.06 ± 0.01% for CN1).

The β-glucan content of the cultivars, as classified by the riceland ecosystem, photoperiod sensitivity, and cooked texture, was also investigated, as shown in Table 2. For each riceland ecosystem, the β-glucan contents of milled rice, in descending order, were 0.11–0.88% for lowland (L) > 0.37–0.71% for upland (U) > 0.21–0.29% for deep-water (D) > 0.28% (only one cultivar) for floating (F). Conversely, for rice bran, the β-glucan contents were 0.25–0.67% for U > 0.11–0.55% for L > 0.29–0.40% for D > 0.33% for F. For rice husk, the β-glucan contents in each of riceland ecosystems were slightly different values in the range of 0.02–0.11% for L > 0.05–0.09% for U > 0.05–0.07% for D > 0.05% for F. The lowland group has the highest number of samples as more than half of Thailand grows lowland rice [1]. Additionally, lowland rice cultivars are almost found in all regions of the country, whereas upland rice is found only in the north and south. Floating rice and deep-water rice cultivars are planted to a lesser extent and found in the eastern region and on the central plain. Therefore, the amount of β-glucan among three fractions from cultivars grown in lowland region was more varied than others. The SN cultivar, which is lowland rice and is grown in northeastern region, showed the highest content for both milled rice and rice husk fractions, which was higher than for the upland JKC cultivar grown in the northern region. This might be due to the northeast of Thailand mostly experiencing a higher temperature, especially Sakon Nakhon province, and lower rainfall than the northern region [43], which is consistent with the study of Howarth et al. [31] that found a higher β-glucan content of grown oats occurred when grain filling in higher temperature and lower rainfall conditions. However, this result is not consistent with bran of the upland SMJ grown in the north, the β-glucan content of which was the highest and slightly higher than that for the upland DPy and lowland SYP grown in the southern region.

Based on the flowering response to light, rice cultivars are grouped into photosensitive and non-photosensitive. Photoperiod-sensitive rice has a longer growing period, resulting in only one crop per year. In contrast, photoperiod-insensitive rice can be grown at any time, even during the dry season when lowland rice areas are being irrigated [14]. In Thailand, the entire northern, northeastern and southern rice-growing regions grow mostly photoperiod-sensitive cultivars [44,45]. The β-glucan contents of milled rice from a group of photoperiod-sensitive (S) cultivars varied from 0.14 to 0.71%; these were slightly lower than those of the group of photoperiod-insensitive (IS) cultivars, which varied from 0.11 to 0.88%. For rice husk, the group of S cultivars in the range of 0.02–0.09% showed slightly lower amounts than the group of IS cultivars in the range of 0.02–0.11%. For rice bran, the group of S cultivars in the range of 0.20–0.67% showed higher amount than the group of IS cultivars in the range of 0.11–0.52%. In the case of milled rice and rice husk, the photoperiod-insensitive rice cultivars growing in the dry season might result in a higher level of β-glucan than the photoperiod-sensitive rice. This is consistent with the finding of Ehrenbergerová et al. [31], which revealed that lower rainfall and warmer weather during flowering time result in an enhanced β-glucan content in barley.

Based on the texture of cooked rice, rice cultivars are grouped into glutinous and non-glutinous rice. The non-glutinous cultivars were grown in all regions of the country, while the glutinous cultivars were mainly grown in the northeast [33]. However, Thai people in the north and northeast prefer glutinous rice. The β-glucan contents in glutinous rice (G) cultivars, which varied from 0.34 to 0.88% for milled rice and 0.23 to 0.67% for rice bran, were higher than those of non-glutinous rice (NG) cultivars, ranging from 0.11 to 0.71% for milled rice and 0.11 to 0.57% for rice bran. For rice husk, G cultivars ranging from 0.05 to 0.11% were relatively similar to NG cultivars ranging from 0.02 to 0.09%. It was found for G cultivars that the β-glucan content of milled rice fraction was higher than that of rice bran, except for the SMJ cultivar. The glutinous rice cultivars with a low amylose content showed a higher β-glucan content than non-glutinous rice due to the amylose content in starchy endosperm being inversely correlated with the β-glucan content, especially in milled rice fraction [33,34,35,36].

The SN cultivar showed the highest β-glucan of all rice fractions in the photoperiod-insensitive and glutinous rice groups, but its rice bran fraction had a lower β-glucan content than SMJ, which is in the same glutinous rice group, but is photoperiod sensitive. In the photoperiod-sensitive and non-glutinous rice groups, JKC also has the highest β-glucan content for all fractions. These results suggest that the milled rice fractions of SN and JKC, as well as the rice bran fraction of SMJ, are better sources of β-glucan than the other cultivars. According to photoperiod sensitivity [14], SN can be grown at any time of year in an irrigated lowland area, but JKC and SMJ can only be grown once a year. 

### 2.3. Descriptive Statistic and Frequency Distribution of β-Glucan Content in Each Rice Fraction

The Kolmogorov–Smirnov test for normality and skewness and kurtosis statistics were applied to the analysis of the β-glucan content of each rice fraction sample. The significant value of the test is 0.05, and an acceptable range for skewness and kurtosis was from +1.96 to −1.96 [46]. The data suggest that β-glucan is normally distributed in rice husk, but not in milled rice or rice bran. Moreover, the kurtosis of 2.732 (SE = 0.449) for milled rice showed that it was not in the acceptable range. This was due to three exceptional cultivars identified by the box plot: SN and JKC in the milled rice population and SMJ for rice bran, which showed a distinctly higher β-glucan content compared to the other cultivars. After removing these three from the milled rice and rice bran populations, the Kolmogorov–Smirnov test suggested that the β-glucan content was normally distributed in both populations with an acceptable skewness range of 0.197 (SE = 0.236) and kurtosis of −0.771 (SE = 0.467) for milled rice. Moreover, the skewness of −0.358 and kurtosis of 0.140 for rice husk and skewness of 0.419 and kurtosis of 0.344 for rice bran (the same SE as milled rice) are presented in Table 3. The mean β-glucan content of milled rice at 0.302% (SE = 0.112, SD = 0.115) was relatively equal to that of rice bran at 0.309% (SE = 0.010, SD = 0.102), whereas rice husk showed the lowest mean β-glucan content at 0.054% (SE = 0.001, SD = 0.015). The β-glucan contents of milled rice with the exceptional SN and JKC cultivars and rice bran with the exceptional SMJ cultivar were similar, ranging from 0.090 to 0.580%. The β-glucan contents of rice husk ranged from 0.010 to 0.080%.

The box plot in Figure 1 illustrates the variation in β-glucan contents among the three fractions. According to the frequency distribution, the majority of the milled rice fraction was between 0.2 and 0.3%, representing 31.6%; the majority of the rice bran fraction was between 0.3 and 0.4%, representing 36.8%; and the majority of the rice husk fraction was between 0.05 and 0.06%, representing 52.6%. These results reveal that the β-glucan content of rice bran was slightly higher than that of milled rice, while little was found in rice husk. This observation is supported by Palmer et al. [28], who found that the aleurone cell walls of rice bran were strongly labeled using MLG antibody, whereas the sub-aleurone and endosperm cells of milled rice were labeled with an even intensity.

### 2.4. Relation of Amylose and β-Glucan Contents in Milled Rice Fraction

To evaluate the relationship between amylose and β-glucan content, amylose in milled rice was determined using an amylopectin/amylose assay kit, and the results are shown in Table 4. Glutinous and non-glutinous rice cultivars were grouped as indicated in Table 2 and classified into four clusters according to the amylose level: 1—waxy (0–6%); 2—very low (>6–14%); 3—low (>14–20%); and 4—intermediate (>20–30%), slightly modified from Juliano [16]. 

Amylose level distribution in milled rice was ranked in the following ascending order: cluster 1 (2.45–5.77%) < cluster 2 (10.07–13.85%) < cluster 3 (15.19–19.97%) < cluster 4 (20.03–27.39%). For the glutinous rice group (cluster 1), RD6 had the lowest amylose content of 2.45 ± 0.08%, close to that of SN with 2.59 ± 0.02%. Glutinous rice RD14 had the highest amylose content of 5.77 ± 0.06%. Among the non-glutinous cultivars, RD41 had the highest amylose content of 27.39 ± 0.25%, close to that for LPt123 with 27.10 ± 0.28%, while JKC had the lowest: 10.07 ± 0.03%. Cluster 1 contained only glutinous rice cultivars, which mostly had less than 6% amylose, whereas clusters 2–4 contained non-glutinous cultivars with amylose in the range of 10–30%. This was consistent with the findings of Chung et al. [15], who stated that glutinous differs from non-glutinous rice mainly in the low amylose content (<5%) in its starch or milled rice.

In contrast to the amylose level, the distribution of β-glucan content was ranked in the following descending order: cluster 1 (0.34–0.88%) > cluster 2 (0.24–0.71%) > cluster 3 (0.21–0.34%) > cluster 4 (0.11–0.18%). This inverse relationship between amylose and β-glucan content was consistent with the findings of Islamovic et al. [33] and lzydorczyk et al. [34], who reported that waxy or low amylose barley had a higher β-glucan level than normal or high-amylose barley. This study showed a similar relationship: the waxy SN cultivar had the highest β-glucan content of 0.88 ± 0.03%, followed by the very low amylose JKC, which had a high β-glucan content of 0.71 ± 0.03%. On the other hand, RD41 had the highest amylose but the lowest β-glucan content of 0.11 ± 0.02%. 

To date, much research has found an inverse relationship between amylose and β-glucan content, especially for barley cultivars, but data for rice are lacking. Hence, the correlation as well as linear regression analysis were studied in the milled rice fraction to confirm the inverse relationship. In the milled rice of all the cultivars, amylose content showed a strong negative correlation with β-glucan (r = −0.805, *p* < 0.0001). In the case of the four clusters, classified by differences in amylose level, the relationship between amylose and β-glucan was separately evaluated, as shown in Table 5 and Figure 2, and showed a negatively moderate to strong relationship (*p* < 0.001). Cluster 2 showed a strong negative correlation (r = −0.857, R^2^ = 0.734), followed by cluster 3, which showed a good negative correlation (r = −0.746, R^2^ =0.556). Clusters 1 and 4, however, showed only a fairly negative correlation (r = −0.622, R^2^ = 0.387 for cluster 1, and r = −0.603, R^2^ = 0.363 for cluster 4). These results exhibited a relationship similar to that observed by Hang et al. [35] and Shu and Rasmussen [36], who found a negative correlation of amylose and β-glucan in barley, reporting a Pearson’s correlation of −0.829 and −0.620 at *p* < 0.01, respectively. The glutinous rice cultivars (cluster 1) revealed a lower correlation of amylose and β-glucan content than for non-glutinous rice cultivars (clusters 2–4). However, the β-glucan level in cluster 1 was higher than for the other clusters. As shown in Figure 2, clusters 1 and 2 of the SN and JKC cultivars, respectively, stand out due to their higher β-glucan content, which is consistent with the findings of the normality test of milled rice β-glucan samples.

## 3. Materials and Methods

### 3.1. Chemicals

The mixed-linkage β-glucan and amylose/amylopectin assay kits were purchased from Megazyme (International Ireland Ltd., Wicklow, Ireland). Ethanol, glacial acetic acid, and dimethyl sulfoxide were purchased from Merck (Darmstadt, Germany). Sodium dihydrogen orthophosphate dihydrate, sodium hydroxide, anhydrous sodium acetate, and sodium chloride were purchased from Qrec (Qrec (Asia) SDN BHD, Selangor, Malaysia) and Ajax Finechem Pty Ltd. (Sydney, New South Wales, Australia). Calcium chloride dihydrate, magnesium chloride hexahydrate, and manganese chloride tetrahydrate were obtained from Sigma-Aldrich Co. (St. Louis, MO, USA). All chemicals and reagents used in the study were of analytical grade.

### 3.2. Rice Sample

The 38 paddy-rice samples in this study were obtained from 12 rice research centers in each province of Thailand. A list of the selected rice cultivars is shown in Table 6: 14 from the northern provinces of Chiang Rai, Chiang Mai, Phrae and Mae Hong Son and the Samoeng Distinct of Chiang Mai province; 6 from Pathum Thani province on the central plain, 4 from the eastern province of Prachinburi; 3 from the western province of Ratchaburi; 2 from the northeastern province of Sakon Nakhon; and 9 from the southern provinces of Nakhon Si Thammarat, Phatthalung and Pattani.

Based on information from the Rice Department at the Ministry of Agriculture and Cooperatives [11], the rice cultivars we studied were categorized according to three factors: riceland ecosystem, photoperiod sensitivity, and cooked texture (Figure 3). The cultivars could also be divided into four groups according to cultivation, namely upland, lowland, floating, and deep-water rice, whereby there were 29 lowland, 5 upland, 3 deep-water, and 1 floating rice cultivars, as listed in Figure 3a. Just over half (23) of all selected cultivars belonged to the group of photoperiod-sensitive rice, whereas 15 were photoperiod insensitive (Figure 3b). According to the texture of cooked white rice, which is influenced by amylose content, 28 cultivars were non-glutinous and 10 were glutinous, as illustrated in Figure 3c.

### 3.3. Yield of Paddy-Rice Fractions

The 38 paddy-rice samples (300 g) were de-hulled and polished using a rice milling machine (Mini-Rice mill Machine, Bangkok, Thailand) to obtain the fractions of milled rice, rice husk, and rice bran, the latter of which includes the rice germ (generally 2.81–3.19% of a rice grain [3]). All fractions were dried at 105 °C for 24 h to a constant weight. Then, the paddy-rice fraction of each sample was weighed and expressed as a percentage of yield according to Equation (1). The milled rice and rice husk were ground. Then, all three fractions were passed through a 0.5 mm sieve before being used for analysis.
Yield of paddy rice fraction (% *w/w*) = (W_RF_/W_R_) × 100 (1)
where W_RF_ is the weight of each paddy-rice fraction (milled rice, rice bran, and rice husk) of the cultivars (g) after rice milling, and W_R_ is the weight of the paddy-rice sample (g).

### 3.4. β-Glucan Content Analysis

A mixed-linkage β-glucan assay for cereal grains kit (K-BGLU, Megazyme, Wicklow, Ireland) was used for the analysis of β-glucan content in each fraction of the 38 rice cultivars. This method uses a highly specific enzyme for cereal mixed-linkage β-glucan hydrolysis without affecting other polysaccharides. To carry out the reaction, 1.0 g of rice fraction sample, after passing through a 0.5 mm sieve screen, was mixed with 5.0 mL of 50% (*v*/*v*) aqueous ethanol and incubated at 100 °C for 5 min. After mixing the sample on a vortexer (VM-300, Gemmy Industrial Corp., Taipei, Taiwan), 5.0 mL of 50% (*v*/*v*) aqueous ethanol was then added, followed by mixing. The sample was then centrifuged at 1000*g* × for 10 min (Z206A, Hermle Labortechnik GmbH, Wehingen, Germany). The pellet was resuspended in 10.0 mL of 50% (*v*/*v*) aqueous ethanol and the centrifugation repeated. The pellet was further mixed and incubated with 5.0 mL of 20 mM sodium phosphate buffer (pH 6.5) (prepared by dissolving 3.12 g of sodium dihydrogen orthophosphate dihydrate in 900 mL of distilled water, adjusting the pH to 6.5 by 100 mM sodium hydroxide, and then adjusting the volume to 1 L) at 100 °C for 2 min, then vigorously stirred and incubated again for 3 min. Then, 0.2 mL of lichenase (10 U, EC 3.2.1.73) was added to the sample and incubated at 40 °C for 1 hr. Distilled water was added to the sample in a tube to adjust the final volume to 30.0 mL, followed by centrifugation at 1000*g* × for 10 min. Furthermore, 3 of 4 supernatant tubes (0.1 mL) were added to 0.1 mL of β-glucosidase (0.2 U, EC 3.2.1.21) plus 50 mM sodium acetate buffer (pH 4.0) (prepared by adding 2.9 mL of glacial acetic acid to 900 mL of distilled water, adjusting the pH to 4.0 by 100 mM sodium hydroxide, and then adjusting the volume to 1 L), while another was added to 0.1 mL of 50 mM sodium acetate buffer (pH 4.0). Then, these samples were incubated at 40 °C for 15 min. Finally, 3.0 mL of GOPOD reagent, containing 36 U of glucose oxidase (EC. 1.1.3.4) plus 1.95 U of peroxidase (EC 1.11.1.7) and 0.24 g of 4-aminoantipyrine was added and incubated at 40 °C for 20 min. An amount of 100 µg of D-glucose standard was added to GOPOD reagent and incubated at 40 °C for 20 min. The β-glucan content was measured at 510 nm using a spectrophotometer (Genesys 20 Spectrophotometer, Thermo Spectronic, New York, NY, USA) and reported on a moisture-free basis. The moisture content was assayed following approved method 44-15.02 [47].

β-glucan content was calculated using the glucose quantity determined with the described assay in Equation (2) and expressed as total β-glucan (mg) in 300 g of paddy-rice using Equation (3).
β-glucan (% *w/w*) = ΔE × (F/mg) × 27(2)
where ΔE is the absorbance difference of the sample after β-d-glucosidase treatment—the absorbance of a blank (non-treatment by β-d-glucosidase); mg is the weight of the sample (mg); and F is a factor for the conversion of absorbance values to µg glucose, which is calculated from 100 (µg of d-glucose)/ absorbance of 100 µg of d-glucose.
Total β-glucan (mg) in 300 g of paddy rice = G × Y × 10(3)
where G is the β-glucan content (% *w*/*w*), and Y is the fraction yield of paddy rice (% *w*/*w*).

### 3.5. Amylose Content Analysis

The amylose content in the milled rice fraction of each rice cultivar was determined using an amylose/amylopectin assay kit (K-AMYL, Megazyme, Wicklow, Ireland). Starch pretreatment was first performed, whereby a 20 mg milled rice sample was completely dispersed in 1.0 mL of dimethyl sulfoxide by heating at 100 °C for 1 min. Then, the mixture was vigorously stirred and incubated at 100 °C for 15 min. After incubating the mixture at room temperature for 5 min, the starch was precipitated out by adding 4.0 mL of 95% (*v*/*v*) aqueous ethanol for 15 min while lipids remaining in the supernatant were removed by centrifugation at 2000*g* × for 10 min. The starch pellet was mixed with 2.0 mL of dimethyl sulfoxide and incubated at 100 °C for 15 min. A measure of 4.0 mL of Con A solvent was immediately added to this mixture. The Con A solvent was freshy prepared by diluting 30 mL of concentrated Con A solvent in 100 mL of distilled water (concentrated Con A solvent prepared by dissolving 49.2 g of anhydrous sodium acetate, 175.5 g of sodium chloride, 0.5 g of calcium chloride dihydrate, 0.7 g of magnesium chloride hexahydrate, and 0.7 g of manganese chloride tetrahydrate in 900 mL of distilled water; adjusting the pH to 6.4 by glacial acetic acid; and then adjusting the volume to 1 L).

Amylopectin specifically forms a complex with lectin concanavalin A (Con A). To the starch suspension, 0.50 mL of Con A solution (4 mg/mL) was added, gently mixed, and allowed to stand at room temperature for 1 h. Then, the amylose supernatant was collected following centrifugation at 4400*g* × for 20 min, and subsequently, the amylopectin pellet was removed. After adding 3.0 mL of 100 mM sodium acetate buffer (pH 4.5) (prepared by adding 5.9 mL of glacial acetic acid to 900 mL of distilled water, adjusting the pH to 4.5 by 1 M sodium hydroxide, and then adjusting the volume to 1 L) to 1.0 mL of the supernatant to decrease the pH to 5.0, the remaining Con A was denatured by heating at 100 °C for 5 min. An amount of 0.1 mL of a mixture of amyloglucosidase (16.5 U, EC 3.2.1.3) and α-amylase (2.5 U, EC 3.2.1.1) enzymes was added to the supernatant, and then incubated at 40 °C for 30 min. After centrifugation at 2000*g* × for 5 min, 1.0 mL of supernatant was mixed with 4.0 mL of GOPOD and incubated at 40 °C for 20 min. The absorbance of Con A supernatant (E_A_) was measured at 510 nm.

Total starch was further determined for the calculation of amylose content. The starch suspension (0.5 mL) was mixed with 4.0 mL of 100 mM sodium acetate buffer (pH 4.5) plus 0.1 mL of amyloglucosidase/α-amylase solution and incubated at 40 °C for 10 min. Then, 1.0 mL of the mixture was added to 4.0 mL of GOPOD reagent and incubated at 40 °C for 20 min. The absorbance of total starch (E_S_) was measured at 510 nm.

The amylose content was estimated using the glucose quantity found in the Equation (4)
Amylose (% *w/w*) = (E_A_/E_S_) × 66.8 (4)
where E_A_ is the absorbance of the Con A supernatant (containing amylose after amylopectin precipitated by lectin concanavalin A) and E_S_ is the absorbance of total starch.

### 3.6. Statistical Analysis

Statistical analyses were conducted using SPSS (v22; IBM, Amonk., New York, NY, USA). The results were calculated as an average of three replicate samples and the results expressed as mean ± standard deviation. A significant difference among each paddy-rice fraction yield, β-glucan, and amylose contents of 38 rice cultivars was determined using the one-way analysis of variance (ANOVA) at the *p* < 0.05 level using the Duncan multiple-range test (DMRT). Pearson coefficient (r) studies were carried out to determine the relationship between amylose and β-glucan contents using bivariate significance at the *p* < 0.01 level.

## 4. Conclusions

In this study, 38 paddy-rice samples grown in six regions of Thailand (northern, central plain, eastern, western, northeastern, and southern) were selected to determine and evaluate the difference in β-glucan contents in three of their fractions (milled rice, rice husk, and rice bran). All the cultivars grown in those regions contained variations in β-glucan levels in each fraction, which were milled rice, rice bran, and rice husk in ranges of 0.11–0.88%, 0.11–0.67%, and 0.02–0.11%, respectively. A slight difference was found in milled rice and rice bran fractions, but both clearly differed from rice husk, which had a low level of β-glucan. Therefore, the β-glucan mainly found in milled rice and rice bran is similar to that found in other cereal grains, generally in the starchy endosperm, sub-aleurone, and aleurone cell walls. The milled rice of the SN cultivar, lowland rice grown in the northeastern region, contained the highest β-glucan content, followed by the milled rice of the JKC cultivar and rice bran of the SMJ cultivar, which are upland rice grown in the northern region. The environmental classification, based on riceland ecosystem and photoperiod sensitivity, showed some patterns with the level of β-glucan content for each paddy-rice fraction, while the cooked texture of glutinous and non-glutinous rice, which is strongly influenced by amylose, was obviously related to the β-glucan content in the milled rice fraction. The significant correlation between β-glucan and amylose content was strongly negative. This research suggests that Thai rice cultivars could be a potential source of beneficial β-glucan, especially in their milled rice and rice bran fractions, for further use as functional ingredients for cereal-based products, supplement products and cosmetics. The application of milled rice of SN and JKC cultivars, as well as rice bran of SMJ, as functional food ingredients could be studied in the future.

## Figures and Tables

**Figure 1 molecules-26-06368-f001:**
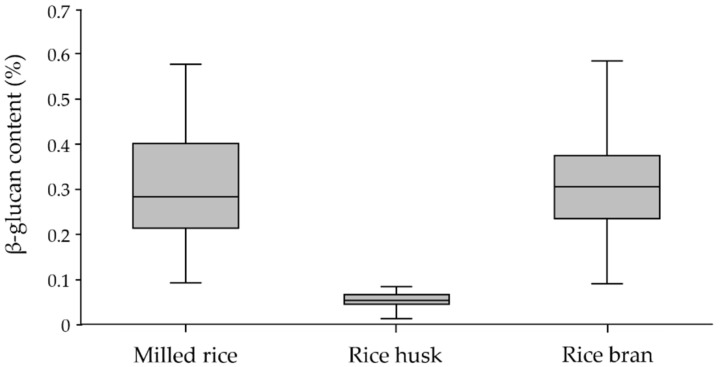
Variation in β-glucan content in milled rice, rice husk, and rice bran fractions of paddy-rice samples with the exceptional SN, JKC, and SMJ cultivars.

**Figure 2 molecules-26-06368-f002:**
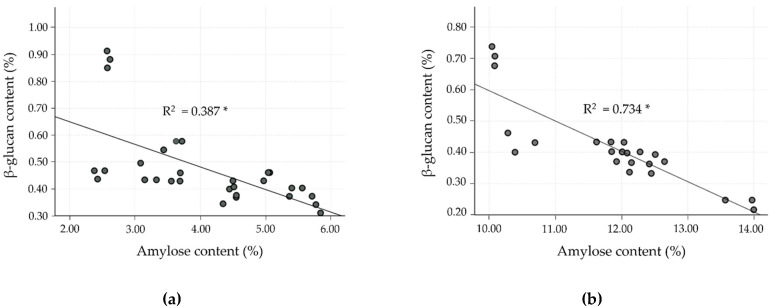
Linear correlation between β-glucan and amylose contents in milled rice fractions of four clusters classified by amylose level. (**a**) Cluster 1: waxy (0–6%); (**b**) cluster 2: very low (>6–14%); (**c**) cluster 3: low (>14–20%); and (**d**) cluster 4: medium (>20–30%). * *p* < 0.0001.

**Figure 3 molecules-26-06368-f003:**
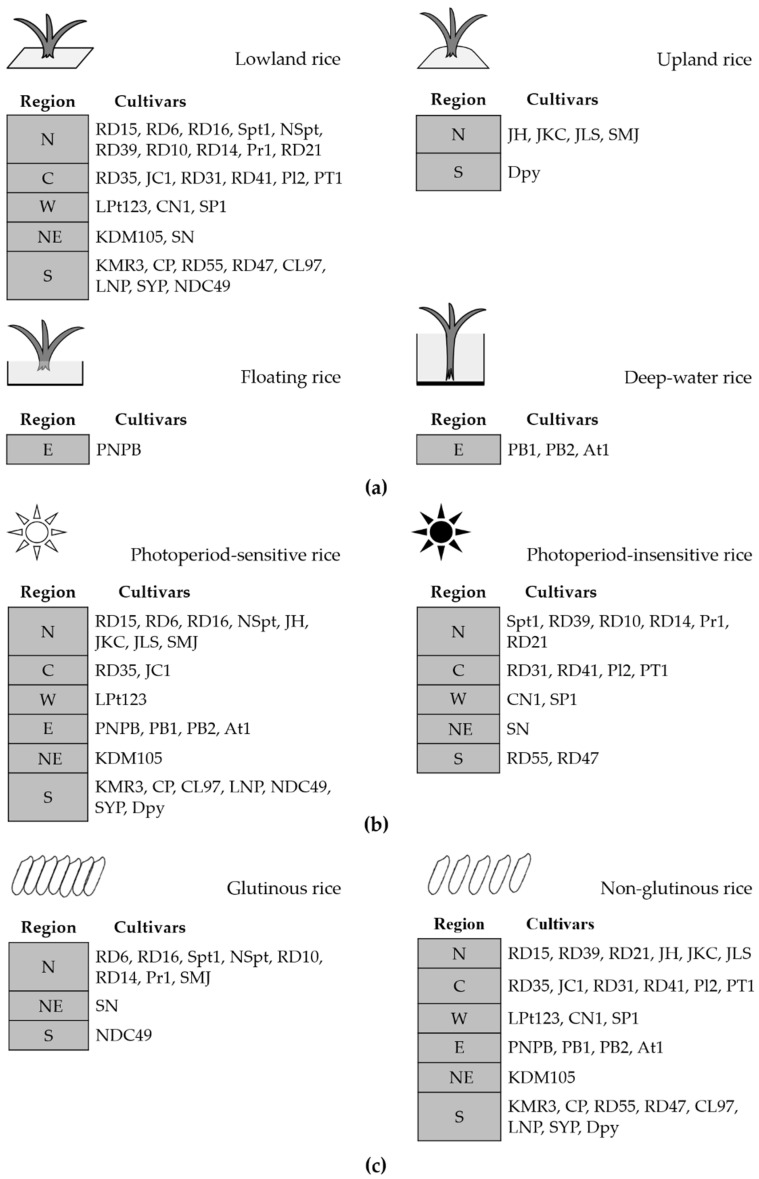
The list of rice cultivars in each region (northern, N; central plain, C; northeastern, NE; eastern, E; western, W; southern, S) categorized according to three factors (**a**) riceland ecosystem; (**b**) photoperiod sensitivity; and (**c**) texture of cooked rice. Abbreviations for the rice cultivars are in Table 1.

**Table 1 molecules-26-06368-t001:** Yield of milled rice, rice husk, and rice bran fractions of the 38 rice cultivars.

No	Cultivar	Yield (% *w*/*w*)			
		Milled Rice	Rice Husk	Rice Bran	Yield Loss *
1	RD15	61.34 ± 0.65 mn	29.09 ± 0.75 ab	8.04 ± 0.42 ijk	1.53 ± 0.27 g
2	RD6	62.30 ± 0.61 kl	27.47 ± 0.22 ef	8.54 ± 0.45 ghij	1.68 ± 0.35 ef
3	RD16	61.67 ± 1.13 mn	27.64 ± 0.78 e	8.98 ± 0.60 fg	1.71 ± 0.24 ef
4	Spt1	64.25 ± 1.10 fgh	28.23 ± 1.09 cd	6.64 ± 0.35 no	0.88 ± 0.14 l
5	NSpt	63.98 ± 1.18 hij	26.74 ± 1.07 ghi	7.48 ± 0.28 lm	1.80 ± 0.29 de
6	RD39	70.08 ± 1.51 a	21.28 ± 1.59 s	7.07 ± 0.27 lmn	1.57 ± 0.36 g
7	RD10	63.13 ± 0.66 jk	25.81 ± 1.01 m	9.54 ± 0.43 bcd	1.52 ± 0.30 gh
8	RD14	62.41 ± 0.65 kl	27.20 ± 1.00 efg	8.57 ± 0.58 ghi	1.82 ± 0.28 de
9	Pr1	62.00 ± 0.32 klm	27.39 ± 0.65 ef	9.18 ± 0.34 def	1.42 ± 0.29 hi
10	RD21	62.14 ± 1.69 klm	29.76 ± 2.19 a	6.26 ± 0.75 o	1.84 ± 0.25 de
11	JH	66.14 ± 1.54 de	24.32 ± 2.33 op	7.43 ± 0.48 lm	2.10 ± 0.37 b
12	JKC	67.27 ± 0.38 c	23.77 ± 0.29 r	7.03 ± 0.21 lmn	1.93 ± 0.34 cd
13	JLS	67.56 ± 0.86 c	23.74 ± 1.19 r	7.42 ± 0.14 lm	1.29 ± 0.21 ij
14	SMJ	65.41 ± 0.38 f	24.05 ± 0.61 opq	9.05 ± 0.32 def	1.49 ± 0.20 gh
15	RD35	61.90 ± 1.39 lmn	27.38 ± 1.06 ef	8.87 ± 0.38 fg	1.85 ± 0.22 de
16	JC1	61.16 ± 0.60 no	28.97 ± 0.55 b	8.67 ± 0.35 fgh	1.21 ± 0.14 k
17	RD31	61.77 ± 3.04 lmn	26.55 ± 1.64 hijk	9.89 ± 0.92 ab	1.79 ± 0.50 de
18	RD41	64.29 ± 1.36 fgh	26.16 ± 0.77 klm	8.05 ± 1.08 ijk	1.50 ± 0.23 gh
19	Pl2	64.75 ± 1.04 fg	26.19 ± 0.49 klm	7.58 ± 0.44 lm	1.48 ± 0.18 gh
20	PT1	61.92 ± 0.68 lmn	28.24 ± 0.54 cd	8.14 ± 0.93 hijk	1.70 ± 0.30 ef
21	LPt123	62.68 ± 0.75 kl	27.57 ± 0.30 e	8.29 ± 0.42 hij	1.46 ± 0.32 gh
22	CN1	62.12 ± 1.94 klm	27.24 ± 2.29 ef	9.17 ± 0.25 def	1.46 ± 0.32 gh
23	SP1	63.25 ± 0.19 ij	26.64 ± 1.62 hij	8.76 ± 1.45 fg	1.35 ± 0.32 hi
24	PNPB	63.22 ± 0.18 ij	27.38 ± 0.29 ef	8.06 ± 0.23 ijk	1.34 ± 0.19 ij
25	PB1	61.52 ± 1.08 mn	27.32 ± 0.59 ef	9.77 ± 0.53 abc	1.39 ± 0.20 hi
26	PB2	64.71 ± 1.22 fg	24.47 ± 0.31 op	8.81 ± 0.46 fg	2.01 ± 0.48 c
27	At1	61.59 ± 1.79 mn	26.89 ± 1.73 gh	10.00 ± 0.92 a	1.52 ± 0.32 gh
28	KDM105	68.22 ± 1.61 b	23.94 ± 1.46 qr	6.56 ± 0.65 no	1.28 ± 0.27 ij
29	SN	66.14 ± 1.28 de	25.30 ± 0.85 mn	6.78 ± 0.33 no	1.78 ± 0.14 de
30	KMR3	66.04 ± 1.05 de	25.39 ± 1.33 mn	6.76 ± 0.62 no	1.81 ± 0.37 de
31	CP	67.16 ± 0.52 cd	24.13 ± 1.11 opq	7.03 ± 0.51 lmn	1.69 ± 0.28 ef
32	RD55	63.64 ± 2.69 ij	26.98 ± 2.30 gh	7.84 ± 0.29 kl	1.54 ± 0.15 g
33	RD47	62.98 ± 2.76 jk	28.58 ± 2.26 bc	6.17 ± 0.47 op	2.27 ± 0.91 a
34	CL97	63.96 ± 1.09 hij	24.65 ± 2.38 ghi	9.46 ± 0.94 bcd	1.93 ± 0.37 cd
35	LNP	62.04 ± 1.10 klm	26.87 ± 1.47 ijk	9.72 ± 0.56 abc	1.37 ± 0.16 hi
36	NDC49	63.39 ± 1.67 ij	26.76 ± 1.07 ghi	8.30 ± 0.63 hij	1.54 ± 0.31 g
37	SYP	62.33 ± 1.05 kl	26.60 ± 0.96 hij	9.22 ± 0.29 bcde	1.85 ± 0.25 de
38	DPy	63.98 ± 0.83 hij	25.42 ± 0.72 mn	9.28 ± 0.30 bcde	1.32 ± 0.19 ij

* Yield loss was calculated by subtracting the percentage of all rice-paddy fractions from one hundred percentage points. The different letters in the same column indicate significant difference (*p* < 0.05) according to Duncan’s multiple range test.

**Table 2 molecules-26-06368-t002:** β-Glucan content and total β-glucan content in each fraction of the 38 rice cultivars obtained from entire regions of Thailand.

Cultivar	Categories			β-Glucan Content (%)			Total β-Glucan Content (mg) ^4^		
	R ^1^	P ^2^	T ^3^	Milled Rice	Rice Husk	Rice Bran	Milled Rice	Rice Husk	Rice Bran
Northern Region									
RD15	L	S	NG	0.34 ± 0.02 lm	0.05 ± 0.01 lmn	0.39 ± 0.01 fg	617.82 ± 28.20 j	41.54 ± 4.94 lm	93.31 ± 2.69 e
RD6	L	S	G	0.46 ± 0.02 d	0.07 ± 0.00 fg	0.31 ± 0.01 jk	854.67 ± 26.14 de	56.45 ± 3.97 de	79.80 ± 1.67 gh
RD16	L	S	G	0.38 ± 0.03 ijkl	0.06 ± 0.00 hij	0.26 ± 0.01 mn	695.32 ± 47.25 ghi	52.20 ± 4.83 fgh	63.33 ± 3.71 i
Spt1	L	IS	G	0.44 ± 0.02 def	0.08 ± 0.01 c	0.23 ± 0.01 op	846.25 ± 19.45 de	67.42 ± 4.04 b	44.92 ± 0.65 l
NSpt	L	S	G	0.45 ± 0.04 de	0.06 ± 0.00 ijk	0.36 ± 0.03 gh	872.17 ± 63.17 d	48.28 ± 1.87 ijk	81.81 ± 4.48 gh
RD39	L	IS	NG	0.24 ± 0.02 qrs	0.05 ± 0.01 lm	0.43 ± 0.02 e	497.90 ± 27.86 klm	31.73 ± 1.83 qr	91.86 ± 0.97 ef
RD10	L	IS	G	0.39 ± 0.02 ij	0.07 ± 0.02 de	0.36 ± 0.02 h	744.35 ± 31.41 fgh	55.22 ± 2.52 defg	102.08 ± 2.09 d
RD14	L	IS	G	0.34 ± 0.03 kl	0.05 ± 0.00 lm	0.31 ± 0.02 jk	639.26 ± 51.44 ij	40.68 ± 4.47 lmn	78.78 ± 1.85 h
Pr1	L	IS	G	0.57 ± 0.02 c	0.06 ± 0.00 jk	0.34 ± 0.02 hi	1054.15 ± 36.40 c	48.11 ± 4.81 ijk	94.62 ± 6.01 e
RD21	L	IS	NG	0.33 ± 0.02 lmn	0.03 ± 0.01 p	0.11 ± 0.01 u	617.98 ± 27.02 j	30.28 ± 3.99 qr	21.12 ± 2.22 n
JH	U	S	NG	0.38 ± 0.02 ijk	0.05 ± 0.01 mn	0.25 ± 0.02 mno	755.23 ± 27.60 fg	34.12 ± 3.17 pqr	56.19 ± 3.20 jk
JKC	U	S	NG	0.71 ± 0.03 b	0.09 ± 0.01 b	0.59 ± 0.01 b	1427.68 ± 56.74 b	67.06 ± 3.37 b	123.57 ± 1.78 c
JLS	U	S	NG	0.43 ± 0.03 fg	0.06 ± 0.01 ijk	0.36 ± 0.01 gh	874.21 ± 71.01 d	42.98 ± 3.25 lm	81.17 ± 1.63 gh
SMJ	U	S	G	0.40 ± 0.03 ij	0.07 ± 0.00 de	0.67 ± 0.03 a	783.76 ± 55.76 ef	52.05 ± 1.30 fghi	182.91 ± 1.37 a
Central Plain									
RD35	L	S	NG	0.14 ± 0.02 uvw	0.02 ± 0.00 q	0.26 ± 0.01 mn	266.32 ± 27.45 qr	20.41 ± 1.55 s	68.10 ± 3.81 i
JC1	L	S	NG	0.27 ± 0.02 opq	0.07 ± 0.01 cde	0.40 ± 0.01 f	489.90 ± 30.27 klm	62.99 ± 4.69 bc	104.49 ± 2.18 d
RD31	L	IS	NG	0.12 ± 0.03 wx	0.06 ± 0.00 hij	0.27 ± 0.02 lm	227.95 ± 46.36 r	49.14 ± 2.55 hij	79.39 ± 4.04 h
RD41	L	IS	NG	0.11 ± 0.02 x	0.05 ± 0.00 lm	0.18 ± 0.02 rs	218.35 ± 30.48 r	39.56 ± 1.02 no	43.15 ± 1.56 lm
Pl2	L	IS	NG	0.18 ± 0.02 uv	0.07 ± 0.00 def	0.34 ± 0.02 hi	341.43 ± 33.29 op	55.85 ± 2.56 def	77.67 ± 2.81 h
PT1	L	IS	NG	0.42 ± 0.02 fgh	0.05 ± 0.00 lmn	0.21 ± 0.02 pq	785.20 ± 24.76 ef	41.31 ± 3.07 lm	51.81 ± 3.20 k
Western Region									
LPt123	L	S	NG	0.16 ± 0.02 uvw	0.04 ± 0.01 o	0.35 ± 0.01 hi	309.55 ± 29.80 pq	35.47 ± 3.86 opq	86.57 ± 2.65 fg
CN1	L	IS	NG	0.21 ± 0.03 rst	0.06 ± 0.01 jk	0.39 ± 0.02 f	397.92 ± 44.74 no	46.32 ± 3.37 jkl	108.33 ± 2.20 d
SP1	L	IS	NG	0.15 ± 0.02 uvw	0.02 ± 0.00 q	0.14 ± 0.02 t	284.76 ± 31.68 pqr	12.90 ± 1.52 t	36.65 ± 6.13 m
Eastern Region									
PNPB	F	S	NG	0.28 ± 0.02 op	0.05 ± 0.00 lmn	0.33 ± 0.01 ij	538.90 ± 34.22 kl	39.48 ± 1.74 no	79.22 ± 3.19 h
PB1	D	S	NG	0.25 ± 0.02 pqr	0.07 ± 0.00 fgh	0.29 ± 0.01 kl	463.18 ± 23.84 mn	55.61 ± 1.08 def	83.62 ± 4.81 gh
PB2	D	S	NG	0.21 ± 0.03 rst	0.05 ± 0.00 h	0.40 ± 0.01 f	405.21 ± 50.48 no	33.50 ± 1.80 qr	105.45 ± 4.56 d
At1	D	S	NG	0.29 ± 0.02 mno	0.06 ± 0.01 hij	0.36 ± 0.01 gh	543.05 ± 17.23 kl	49.65 ± 3.63 ghij	108.15 ± 6.98 d
Northeastern Region									
KDM105	L	S	NG	0.41 ± 0.02 hi	0.06 ± 0.00 ghi	0.20 ± 0.02 qr	842.02 ± 18.98 de	46.11 ± 0.97 jkl	38.36 ± 0.70 ml
SN	L	IS	G	0.88 ± 0.03 a	0.11 ± 0.01 a	0.52 ± 0.02 d	1749.21 ± 29.62 a	81.62 ± 2.36 a	105.94 ± 1.86 d
Southern Region									
KMR3	L	S	NG	0.24 ± 0.03 qrs	0.04 ± 0.01 o	0.30 ± 0.01 k	482.47 ± 52.81 lm	33.03 ± 4.21 qr	60.70 ± 3.47 j
CP	L	S	NG	0.20 ± 0.02 stu	0.04 ± 0.00 op	0.25 ± 0.02 mno	412.60 ± 33.55 no	28.54 ± 1.92 r	52.35 ± 3.72 k
RD55	L	IS	NG	0.18 ± 0.03 uv	0.04 ± 0.01 o	0.23 ± 0.02 nop	347.77 ± 47.63 op	34.08 ± 3.45 pqr	54.43 ± 6.16 jk
RD47	L	IS	NG	0.22 ± 0.03 rst	0.05 ± 0.01 h	0.23 ± 0.02 op	406.65 ± 40.62 no	39.10 ± 1.75 nop	41.92 ± 0.48 lm
CL97	L	S	NG	0.29 ± 0.03 no	0.02 ± 0.00 mn	0.28 ± 0.02 kl	558.42 ± 76.60 k	34.12 ± 3.21 pqr	80.46 ± 3.76 gh
LNP	L	S	NG	0.26 ± 0.02 opq	0.06 ± 0.01 hij	0.41 ± 0.01 f	486.25 ± 35.03 klm	50.19 ± 4.47 ghi	118.61 ± 6.47 c
NDC49	L	S	G	0.45 ± 0.02 de	0.05 ± 0.00 l	0.30 ± 0.02 st	856.39 ± 12.04 de	42.57 ± 2.78 klm	39.68 ± 2.53 lm
SYP	L	S	NG	0.36 ± 0.03 jkl	0.06 ± 0.01 jk	0.55 ± 0.01 c	677.82 ± 46.34 hij	43.89 ± 2.75 klm	152.96 ± 7.08 b
DPy	U	S	NG	0.37 ± 0.03 ijkl	0.08 ± 0.00 cd	0.57 ± 0.02 bc	704.51 ± 53.97 c	59.20 ± 2.59 cd	157.73 ± 3.35 b

^1^ Riceland ecosystem: lowland rice, L; upland rice, U; floating rice, F; deep-water rice, D. ^2^ Photoperiod sensitivity: sensitive, S; insensitive, IS. ^3^ Texture of cooked rice: glutinous rice, G; non-glutinous rice, NG. ^4^ Total β-glucan content of each fraction in 300 g of paddy rice. Different letters in the same column mean significantly different (*p* < 0.05) according to Duncan’s multiple range test.

**Table 3 molecules-26-06368-t003:** Descriptive statistics for the β-glucan content of the milled rice, rice husk, and rice bran fractions of paddy-rice samples with the exceptional SN, JKC, and SMJ cultivars.

Descriptive Statistics	Fraction of Paddy Rice		
	Milled Rice	Rice Husk	Rice Bran
Mean (%)	0.302	0.054	0.309
Standard Error of Mean	0.112	0.001	0.010
Variance	0.013	0.000	0.011
Standard Deviation	0.115	0.015	0.102
Skewness	0.197	−0.358	0.419
Standard Error of Skewness	0.236	0.236	0.236
Kurtosis	−0.771	0.140	0.344
Standard Error of Kurtosis	0.467	0.467	0.467
Range (%)	0.480	0.070	0.490
Minimum (%)	0.090	0.010	0.090
Maximum (%)	0.580	0.080	0.580

**Table 4 molecules-26-06368-t004:** Amylose contents in milled rice fractions of four rice cultivar clusters.

Cluster 1 ^1^		Cluster 2 ^2^		Cluster 3 ^3^		Cluster 4 ^4^	
G ^4^ Cultivar	Amylose Content (%)	NG ^5^ Cultivar	Amylose Content (%)	NG ^5^ Cultivar	Amylose Content (%)	NG ^5^ Cultivar	Amylose Content (%)
RD6	2.45 ± 0.08 y	JKC	10.07 ± 0.03 r	RD15	15.19 ± 0.10 l	RD55	20.03 ± 0.12 f
SN	2.59 ± 0.02 y	JLS	10.45 ± 0.22 q	RD21	16.50 ± 0.23 k	Pl2	22.69 ± 0.21 e
NSpt	3.19 ± 0.12 x	PT1	11.77 ± 0.03 p	CL97	17.30 ± 0.08 j	CN1	22.76 ± 0.12 e
Pr1	3.60 ± 0.14 e	KDM105	12.11 ± 0.15 o	LNP	17.41 ± 0.20 j	CP	24.38 ± 0.23 d
Spt1	3.65 ± 0.08 w	DPy	12.12 ± 0.03 o	JC1	17.75 ± 0.09 i	RD35	26.42 ± 0.19 c
RD16	4.47 ± 0.11 v	JH	12.20 ± 0.40 no	KMR3	17.87 ± 0.22 i	RD31	26.54 ± 0.27 c
SMJ	4.50 ± 0.05 v	SYP	12.46 ± 0.05 n	At1	18.58 ± 0.24 h	SP1	26.63 ± 0.18 c
NDC49	5.02 ± 0.05 u	RD39	13.85 ± 0.24 m	PNPB	18.69 ± 0.11 h	LPt123	27.10 ± 0.28 b
RD10	5.44 ± 0.10 t			PB1	19.22 ± 0.09 g	RD41	27.39 ± 0.25 a
RD14	5.77 ± 0.06 s			RD47	19.48 ± 0.26 g		
				PB2	19.97 ± 0.23 f		

^1^ 0–6% amylose (waxy) ^2^ >6–14% amylose (very low), ^3^ >14–20% amylose (low) ^4^ >20–30% of amylose (intermediate); ^4^ glutinous rice, G; ^5^ non-glutinous rice, NG. Mean with the different letters are significantly different (*p* < 0.05) by Duncan’s multiple range test.

**Table 5 molecules-26-06368-t005:** Pearson’s correlation coefficients, linear equation, R^2^, sample sizes, and *p*-value for amylose and β-glucan contents in milled rice fraction.

Cluster	Pearson’s Correlation	Linear Equation	R^2^	Sample Size	*p*-Value
1	−0.622	y = 0.819 − 0.084x	0.387	30	<0.0001
2	−0.857	y = 1.565 − 0.097x	0.734	24	<0.0001
3	−0.746	y = 0.712 − 0.024x	0.556	33	<0.0001
4	−0.603	y = 0.395 − 0.009x	0.363	27	<0.0001

**Table 6 molecules-26-06368-t006:** List of the 38 paddy-rice cultivars from various rice research centers.

No.	Rice Cultivar	Abbreviation	Location of Rice Research Center
1	RD15	RD15	Chiang Rai
2	RD6	RD6	Chiang Rai
3	RD16	RD16	Chiang Rai
4	San-pah-tawng 1	Spt1	Chiang Mai
5	Niaw San-pah-tawng	NSpt	Chiang Mai
6	RD39	RD39	Chiang Mai
7	RD10	RD10	Phrae
8	RD14	RD14	Phrae
9	Phrae 1	Pr1	Phrae
10	RD21	RD21	Mae Hong Son
11	Jow Haw	JH	Mae Hong Son
12	Jow Khao Chiangmai	JKC	Mae Hong Son
13	Jow Lisaw San-pah-tawng	JLS	Mae Hong Son
14	Sew Mae Jan	SMJ	Samoeng
15	RD35 (Rangsit 80)	RD35	Pathum Thani
16	Jek Chuey 1	JC1	Pathum Thani
17	RD31 (Pathum Thani 80)	RD31	Pathum Thani
18	RD41	RD41	Pathum Thani
19	Phitsanulok 2	Pl2	Pathum Thani
20	Pathum Thani 1	PT1	Pathum Thani
21	Leuang Pratew 123	LPt123	Ratchaburi
22	Chai Nat 1	CN1	Ratchaburi
23	Suphan Buri 1	SP1	Ratchaburi
24	Plai Ngahm Prachin Buri	PNPB	Prachinburi
25	Prachin Buri 1	PB1	Prachinburi
26	Prachin Buri 2	PB2	Prachinburi
27	Ayutthaya 1	At1	Prachinburi
28	Khao Dawk Mali 105	KDM105	Sakon Nakhon
29	Sakon Nakhon	SN	Sakon Nakhon
30	Khai Mod Rin 3	KMR3	Nakhon Si Thammarat
31	Chiang Phatthalung	CP	Phatthalung
32	RD55	RD55	Phatthalung
33	RD47	RD47	Phatthalung
34	Cho Lung 97	CL97	Pattani
35	Leb Nok Pattani	LNP	Pattani
36	Niaw Dam Chaw Mai Pai 49	NDC49	Pattani
37	Sang Yod Phattalung	SYP	Phatthalung
38	Dawk Pa-yawm	DPy	Phatthalung

## Data Availability

Not applicable.

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
