# Peer review of "Comparison of β-Glucan Content in Milled Rice, Rice Husk and Rice Bran from Rice Cultivars Grown in Different Locations of Thailand and the Relationship between β-Glucan and Amylose Contents"

_molecules, 2021, doi:10.3390/molecules26216368_

Round 1

Reviewer 1 Report

The manuscript entitled “Comparison of β-Glucan Content in Milled Rice, Rice Husk and Rice Bran from Rice Cultivars grown in Different Locations of Thailand and the Relationship between β-Glucan and Amylose Contents” This work is merit for publication at Molecules after some major modification. So I have some points that may help to improve the work as follows:

1-Abstract is good but need more explain about the main aim of work

2- The introduction should be extended to discuss the hypothesis and research questions in details. Additionally, the introduction should cover the recent literature related to this subject.

3- Material and methods

The methodologies should be explained in details so that the results are reproducible.

4-Results

The results are clear and important.

5-Discussion
The discussion section still needs improvement, and should be linked to the findings of the previous reports on this topic.

5- The conclusion

A section for conclusions need more explain and should include the most significant findings and future works only.

6- English writing should be checked by a native English speaking expert.

Author Response

  • Comment 1: Abstract is good but need more explain about the main aim of work

Response: Agree. I have, accordingly added more explain about the main aim of work. However, the abstract is limit word max 200, so some words were deleted. I have revised as following.

β-glucan is a dietary fiber that is beneficial to human health, and its content varies according to its different parts, type of cereal grain and growing environment. In this study, the β-glucan of milled rice, rice husk and rice bran fractions, as well as the amylose content of milled rice fraction, from 38 selected rice-paddy grains from six regions of Thailand were quantitatively determined. The milled rice of the Sakon Nakhon (SN) cultivar grown in the northeast contained the highest β-glucan content (0.88 ± 0.03%), followed by the milled rice of the Jow Khao Chiangmai (JKC) cultivar (0.71 ± 0.03%) and rice bran of the Sew Mae Jan (SMJ) cultivar (0.67 ± 0.03%) grown in the north. The results reveal that the rice cultivars from each region showing variation in the β-glucan level in each fraction, which is mainly found in milled rice and rice bran, are similar to those found in other cereal grains, although low amounts are found in the husk. The amylose and β-glucan contents in the milled rice fraction showed a strong negative correlation (r = −0.805; p < 0.0001). This new information about the β-glucan content of Thai rice cultivars could be used for the development of cereal-based functional food.

  • Comment 2: The introduction should be extended to discuss the hypothesis and research questions in details. Additionally, the introduction should cover the recent literature related to this subject.

Response: Thank you for pointing this out. I agree with this comment. I have, accordingly, extended to discuss the hypothesis and research details and added more recent literature related as following in line 83-88.

The β-glucan level of cereal can vary due to the nature of the genotype, growing environ-ment, including conditions of flowering time and grain filling. Many researchers have studied oats and barley, with findings such as drier and warmer temperatures during the flowering time of barley could enhance the level of beta-glucan [31], whereas a higher temperature and lower rainfall during oat grain filling contributed to a higher level of be-ta-glucan [32].

Two recent literatures related to the subject as following:

  1. Ehrenbergerová, J.; Brezinová Belcredi, N.; Psota, V.; Hrstková, P.; Cerkal, R.; Newman, C.W. Changes caused by genotype and environmental conditions in beta-glucan content of spring barley for dietetically beneficial human nutrition. Plant Foods Hum. Nutr., 2008, 63, 111-117.
  2. Howarth, C.J.; Martinez-Martin, P.M.J.; Cowan, A.A.; Griffiths, I.M.; Sanderson, R.; Lister, S.J.; Langdon, T.; Clarke, S.; Fradgley, N.; Marshall, A.H. Genotype and environment affect the grain quality and yield of winter oats (Avena sativa L.). Foods 2021, 10, 2356.
  • Comment 3: Material and methods

The methodologies should be explained in details so that the results are reproducible.

Response: Thank you for pointing this out. I agree with this comment. I think that some details about buffer preparation in both methodology of β-glucan and amylose contents determination and the detail of enzyme used are not enough.  Therefore, I have added more details in some methodologies that would be support to reproduce the results, as following: added more detail about buffer preparation in Line 391-393, 398-400, 429-433, 438-440.

                      : added EC code for all enzymes used in Line 394, 398, 402, 403, 442, 443.

  • Comment 4: Results

The results are clear and important.

Response: Many thanks for giving me this opportunity.

  • Comment 5: Discussion

The discussion section still needs improvement, and should be linked to the findings of the previous reports on this topic.

Response: Thank you for pointing this out. I agree with this comment. I think that the discussion section still need improvement and should be linked to the finding of the previous report, you mean as following point that I have revised in line 177-227, page 6.

  • Comment 6: The conclusion

A section for conclusions need more explain and should include the most significant findings and future works only.

Response: Thank you for pointing this out. I agree with this comment. I have added more explain and include the most significant findings and future works in the conclusion as following:

In this study, 38 paddy-rice samples grown in six regions of Thailand (northern, central plain, eastern, western, northeastern and southern) were selected to determine and evaluate the difference in β-glucan contents in three of their fractions (milled rice, rice husk and rice bran). All the cultivars grown in those regions contained variations in β-glucan levels in each fraction, which were milled rice, rice bran and rice husk in ranges of 0.11–0.88%, 0.11–0.67% and 0.02–0.11%, respectively. A slight difference was found in milled rice and rice bran fractions, but both clearly differed from rice husk, which had a low level of β-glucan. Therefore, the β-glucan mainly found in milled rice and rice bran is similar to that found in other cereal grains, generally in the starchy endosperm, sub-aleurone and aleurone cell walls. The milled rice of the SN cultivar, lowland rice grown in the northeastern region, contained the highest β-glucan content, followed by the milled rice of the JKC cultivar and rice bran of the SMJ cultivar, which are upland rice grown in the northern region. The environmental classification, based on riceland ecosystem and photoperiod sensitivity, showed some patterns with the level of β-glucan content for each paddy-rice fraction, while the cooked texture of glutinous and non-glutinous rice, which is strongly influenced by amylose, was obviously related to the β-glucan content in the milled rice fraction. The significant correlation between β-glucan and amylose content was strongly negative. This research suggests that Thai rice cultivars could be a potential source of beneficial β-glucan, especially in their milled rice and rice bran fractions, for further use as functional ingredients for cereal-based products, supplement products and cosmetics. The application of milled rice of SN and JKC cultivars, as well as rice bran of SMJ, as functional food ingredients could be studied in the future.

  • Comment 7: English writing should be checked by a native English speaking expert.

Response: Thank you for pointing this out. I agree with this comment. Therefore, I have edited the English writing in my manuscripts) by English Editing Services by MDPI (english-35317).

Reviewer 2 Report

Authors should review Table 2 and indicate the meaning of the letters that appear in each column next to the% performance.
The same happens in table 3.
In table 4, the units of measure must appear.  

Author Response

  • Comment 1: Authors should review Table 2 and indicate the meaning of the letters that appear in each column next to the% performance.

Response: Thank you for pointing this out. I agree with this comment. Therefore, I have revised the review of Table 2 to “The percentage yields of rice paddy fractions that is milled rice, rice husk and rice bran, and the yield loss of all 38 cultivars are given in Table 2.” in line 108-109.

The letters that appear in each column next to the % is statistical data obtaining by using the one-way analysis of variance (ANOVA) at p < 0.05 level using the Duncan multiple-range test (DMRT). I have already indicated in the below of Table 2 that is “The different letters in the same column mean significantly different (p < 0.05) according to Duncan’s multiple range test.”           

  • Comment 2: The same happens in table 3.

Response: Thank you for pointing this out. I agree with this comment. Therefore, I have revised the review of Table 3 to “The β-glucan content and total β-glucan content of milled rice, rice husk and rice bran fractions in 38 rice varieties that are cultivated in different locations of Thailand and categorized according to riceland ecosystem, photoperiod sensitivity and texture of cooked rice are shown in Table 3.” in line 135-138.

The letters that appear in each column next to the % is statistical data obtaining by using the one-way analysis of variance (ANOVA) at p < 0.05 level using the Duncan multiple-range test (DMRT). I have already indicated in the below of Table 3 that is “The different letters in the same column mean significantly different (p < 0.05) according to Duncan’s multiple range test.”

  • Comment 3: In table 4, the units of measure must appear.

Response: Agree. I have, accordingly, revised in Table 4.

Reviewer 3 Report

In the paper entitled “Comparison of β-Glucan Content in Milled Rice, Rice Husk and Rice Bran from Rice Cultivars grown in Different Locations of Thailand and the Relationship between β-Glucan and Amylose Contents”, the author offers a quite wide investigation of the β-glucan content in rice from Thailand. In particular, different cultivars set in different regions of Thailand have been tested and different types of rice (e.g., milled rice, rice husk and rice bran) have been subjected to analysis.

The work has been carried out accurately and offers also the inspiration for a discussion on how the environmental conditions can modulate the nutritional content of rice. Despite the great effort in performing the all the scientific work, in my opinion Molecules is not the appropriate journal for this kind of research because the topic does not fit with the scope of the journal.

For this reason I reject this paper but I strongly encourage the author in selecting a more suitable journal.

Author Response

  • Comment 1: In the paper entitled “Comparison of β-Glucan Content in Milled Rice, Rice Husk and Rice Bran from Rice Cultivars grown in Different Locations of Thailand and the Relationship between β-Glucan and Amylose Contents”, the author offers a quite wide investigation of the β-glucan content in rice from Thailand. In particular, different cultivars set in different regions of Thailand have been tested and different types of rice (e.g., milled rice, rice husk and rice bran) have been subjected to analysis.

The work has been carried out accurately and offers also the inspiration for a discussion on how the environmental conditions can modulate the nutritional content of rice. Despite the great effort in performing the all the scientific work, in my opinion Molecules is not the appropriate journal for this kind of research because the topic does not fit with the scope of the journal.

For this reason I reject this paper but I strongly encourage the author in selecting a more suitable journal.

Response: Thank you for this suggestion. You have raised an important point here. However, I believe that my manuscript would be in the scope of chemical biology in Molecule Journal. Especially, the quantitative determination of the β-glucan content of rice paddy fractions and amylose content of milled rice by using enzymatic method.

Author Response

Thank your very much for evaluation and the thorough checking for the text. We tried to do my very best to remove all the ambiguities which might have occurred. Of course, because of the changes taking place, the numbering of lines was altered.

Round 2

Reviewer 1 Report

The authors have made changes to the manuscript, so I consider it can be accepted for publication.

Reviewer 3 Report

I appreciated the revision work made by the author. Anyway, the focus of the paper is not the development of a new analytical method for assessing the β-glucan content, no comparison of the proposed enzymatic method with other reported in the literature is discussed. The main focus is on the nutritional impact from rice of different cultivars. In my opinion this paper should be submitted to a jopurnal dealing with food or supplement topic.